

# Enhanced Vegetation Evapotranspiration Increases Precipitation in Oasis Regions

Yinying Jiao[1,2,3], Guofeng Zhu[1,2,3*], Yuxin Miao[1,2,3],Yani Gun[1,2,3], Jiangwei Yang[1,2,3], Qinqin Wang[1,2,3]

[1]College of Geography and Environment Science, Northwest Normal University, Lanzhou 730070, Gansu, China

[2]Shiyang River Ecological Environment Observation Station, Northwest Normal University, Lanzhou 730070, Gansu, China

[3]Key Laboratory of Resource Environment and Sustainable Development of Oasis, Gansu Province, Lanzhou 730070, Gansu, China

*Correspondence to*: Guofeng Zhu (zhugf@nwnu.edu.cn)

**Abstract.** While the impact of vegetation on global climate has been confirmed, the feedback mechanisms between vegetation and precipitation at local scales remain unknown. This study selects oasis as relatively independent geographical units and analyzes stable isotopes in precipitation, soil water, and xylem water across four different vegetation cover areas. Results show that in oasis areas, tree-covered regions have the highest recycled vapor ratio ($f_{re}$), averaging 53%, and the lowest raindrop re-evaporation rate ($f_{re-ev}$) at 38%. Cropland, grassland, and shrub-covered areas have lower $f_{re}$ (39%) and higher average $f_{re-ev}$ (between 60-70%). In comparison, desert areas show more extreme differences between these two vapor ratios, further indicating that vegetation transpiration can increase precipitation by inhibiting sub-cloud re-evaporation loss of raindrops. This provides new insights into the impact of local vegetation on precipitation changes. In future assessments of water resources in arid environments, the effects of vegetation transpiration, recycled vapor, and secondary evaporation of precipitation on local water resources cannot be ignored.

## 1 Introduction

Globally, species better adapted to warmer environments are gaining an advantage. In recent decades, many temperate forests have seen continuous expansion of greening areas and gradual increases in tree biomass, resulting in a canopy closure effect that buffers the impact of atmospheric warming on understory vegetation (De et al., 2013). Since the last glacial period, numerous terrestrial sediment records have shown characteristics of rapid climate change (Peteet, 2000), including severe extreme climate events, leaving irreversible traces on Earth's ecosystems. To mitigate the effects of climate change, plants have increased their productivity and ecosystem stability, and enhanced the stability of temperature and precipitation by acting as a buffer zone (Huang et al., 2024).

The growth of vegetation is a result of the synergistic stimulation from increased water and nutrient supply under global warming. Prolonged water scarcity can determine the sensitivity of biological communities to drought, leading to rapid vegetation responses to adapt to changing water resource mechanisms in short-term drought scenarios (Li et al., 2023). In the global hydrological cycle, vegetation acts as a biological pump, continuously regulating the dynamic balance of water vapor



in the atmosphere. Evaporation and cooling are key factors driven by vegetation that link surface energy redistribution with terrestrial carbon cycling processes (Wang and Zeng, 2024). Changes in evapotranspiration may alter cloud formation and subsequent rainfall events (Zhang et al., 2023). Since rainfall generation requires water vapor transport from the surface and advective directions, insufficient water vapor transport through evapotranspiration will greatly increase the probability of drought occurrences (Gimeno-Sotelo et al., 2024).

The impact of vegetation type on atmospheric water deficit is crucial for solving the problem of precipitation distribution patterns, and it is also essential for understanding the dynamic balance of various components of terrestrial ecosystems (Vicente-Serrano et al., 2013). As a humid region on the edge of the desert, the special geographical location of the oasis not only has important ecological service value but also greatly promotes the social and economic development of the region. Oasis vegetation has typical drought-tolerant and salt-resistant abilities, mainly divided into three categories: trees, shrubs, and grasses. Moreover, the relatively abundant water resources provide favorable conditions for the development of irrigated agriculture in the region. In this study, we use stable isotopes of water to determine the sensitivity of precipitation to changes in evapotranspiration of different vegetation types and to address the following issues: 1) Quantifying the evapotranspiration ratios of forest, farmland, grassland, and shrub ecosystems in arid oasis areas; 2) Determining the impact of local water vapor recycling on precipitation. The research results will help fully elucidate the impact of vegetation evapotranspiration on precipitation and consider the regulatory effect of vegetation on precipitation and atmospheric water vapor from the perspective of heat transfer.

## 2 Study Area

The Shiyang River Basin is located in the heart of the Eurasian continent (38°25′—39°17′N, 102°45′—104°15′E) (Figure 1), with the upstream basin extending between the Badain Jaran Desert and the Tengger Desert, where oases are widely distributed. The middle and lower reaches of the basin have an elevation between 1440-2100m, with lacustrine, alluvial, and proluvial plains as the typical geomorphological features. The area has a continental warm temperate climate due to its distance from the ocean, characterized by intense solar radiation, significant annual and diurnal temperature fluctuations, brief but hot summers, and prolonged, severe winters. The annual precipitation (113.2mm) is significantly less than the annual evaporation (2580.7mm), and wind erosion is severe. Water resources can be broadly divided into surface water and groundwater, with precipitation contributing minimally to local water supply. The main oasis soil types include gray-brown desert soil, marsh soil, saline soil, and meadow soil. The vegetation includes agricultural ecosystems and natural vegetation landscapes, with drought-resistant willow as the main dominant tree species, Russian olive and saxaul as the main dominant shrub species, and reed as the main dominant herbaceous species.





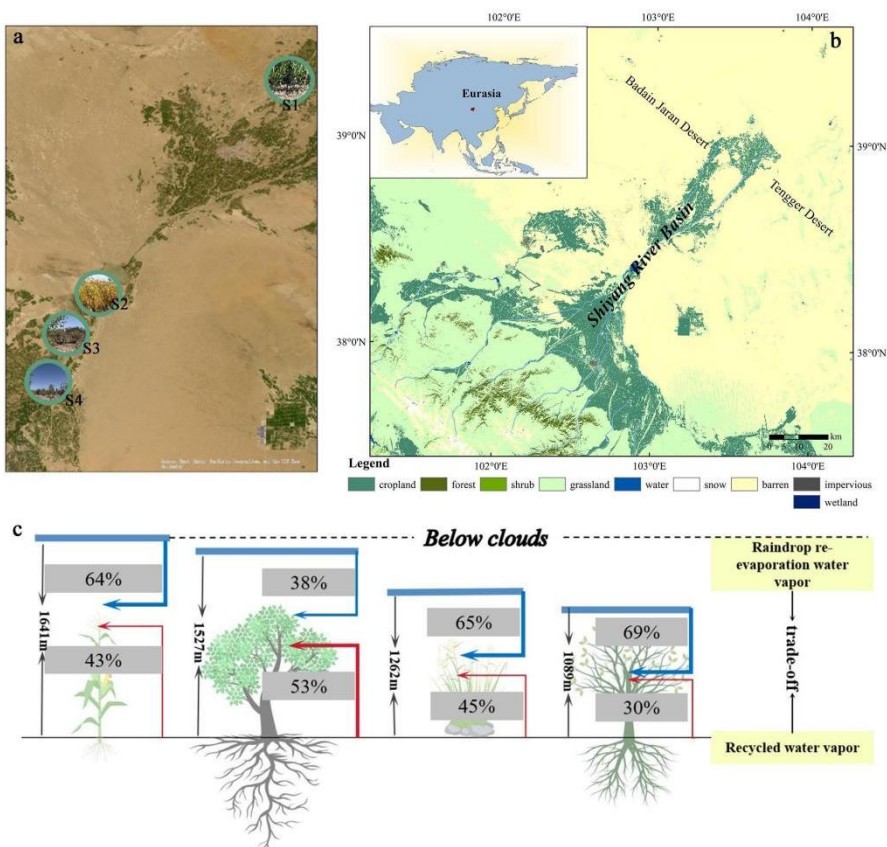


**Figure 1: Overview of the study area. a Sampling point locations and vegetation types, b Geographic location of the study area, c Complementary relationship between recycled water vapor and secondary evaporation vapor under rain clouds (from © Google Maps).**

## 3 Materials and Methods

### 3.1 Sample collection


From April 2018 to October 2022, samples of crop stalks, herbaceous plants, shrubs, and tree stems and lateral branches, as well as soil samples from 0 to 100 cm depth, were collected monthly at sampling points S1, S2, S3, and S4 in the middle and lower reaches of the Shiyang River Basin (SI Appendix, Fig S1). After each precipitation event, samples were collected using 80ml graduated polyethylene bottles. All collected samples were refrigerated for preservation. Automatic weather

stations were set up near each sampling point to collect basic meteorological data including temperature, relative humidity, and atmospheric pressure during the study period. Additionally, daily precipitation and evaporation data from the Wuwei Shiyang River Experimental Station were collected for the period from January 1, 2020, to December 31, 2021.



## 3.2 Experimental Analysis

Before conducting the experimental analysis, all samples were coded based on their collection site and time. Subsequently,
soil and xylem samples were transferred to 10ml glass vials, with cotton wool inserted at the vial openings to minimize
evaporation. Water was then extracted using a low-temperature vacuum distillation system (LI-2100, LICA United
Technology Limited, China). The extracted soil water, xylem water, and precipitation samples were transferred to 2ml glass
vials, the hydrogen and oxygen isotope values of each water sample were then determined using a liquid water isotope
analyzer (DLT-100, Los Gatos Research, USA). The results were expressed using the delta notation "δ" in parts per thousand.

## 3.3 Research Methods

### 3.3.1 Craig and Gordon Model

A binary linear mixing model was used to estimate the ratio (fT) of transpiration (T) to total ecosystem evapotranspiration
(ET):

$$f_T = \frac{\delta_{ET} - \delta_E}{\delta_T - \delta_E} \times 100 \,, \tag{1}$$

Given the substantial differences in oxygen isotope composition between soil and vegetation water, and the significant
depletion of water vapor during evaporation (Dubbert et al., 2013), the Craig and Gordon (1965) model was employed to
determine the oxygen isotope composition of soil evaporation ($\delta_E$) and stem water enrichment ($\delta_x$):

$$\delta_E = \frac{\alpha^+ \delta_{soil} - h\delta_a - \varepsilon^+ - (1-h)\varepsilon_k}{(1-h) + 10^{-3}(1-h)\varepsilon_k}, \tag{2}$$

$$\delta_a = (\delta_P - \varepsilon^+)/\alpha^+, \tag{3}$$

Where $\delta_{soil}$ denotes the oxygen isotope value of soil water at 0-5cm depth, and δa represents the oxygen isotope value of
atmospheric water vapor.

Assuming that in vegetation transpiration, the oxygen isotope composition of water vapor leaving the leaves ($\delta_x$) is identical
to that of xylem water ($\delta_T$) (Han et al., 2022), the following equation holds:

$$\delta_T \approx \delta_x, \tag{4}$$

Based on the fitting equation of atmospheric water vapor concentration (Ca) and stable isotopes (δa), the Keeling plot
equation is obtained (Keeling, 1958; Wang et al., 2015), with the intercept representing the ecosystem's isotope value ($\delta_{ET}$).
The equation is as follows:

$$\delta_a = \frac{1}{C_a}[C_{bg}(\delta_{bg} - \delta_{ET})] + \delta_{ET}, \tag{5}$$

Where $C_{bg}$ and $\delta_{bg}$ denote the water vapor concentration and isotope value of the background atmosphere, respectively.





### 3.3.2 Bayesian Mixing Model

The water vapor in precipitation primarily comes from surface evapotranspiration and external water vapor transport (Burde & Zangvil, 2001). The contribution of recycled moisture to precipitation can be represented as:

$$\delta_{pv} = \delta_t f_t + \delta_{ev} f_{ev} + \delta_{adv} f_{adv}, \tag{6}$$

$$f_t + f_{ev} + f_{adv} = 1, \tag{7}$$

The stable isotope composition of precipitation water vapor (δpv) is computed using the following equation:

$$\delta_{pv} = \frac{\delta_p - k\varepsilon^+}{1 + k\varepsilon^+}, \tag{8}$$

Where δp represents the stable isotope composition of precipitation, k is an adjustment parameter, $\delta^{18}O_{ev}$ is calculated using the C-G model, with the formula as follows:

$$\delta_{ev} = \frac{\delta_s/\alpha^+ - h\delta_{adv} - \varepsilon}{1 - h + \varepsilon_k}, \tag{9}$$

Where $\delta_{ev}$ is the isotope composition of soil evaporation water vapor, $\delta_{adv}$ is the isotope composition of advected water vapor, h is the relative humidity, $\alpha^+$ is the equilibrium fractionation factor, $\varepsilon_k$ is the kinetic fractionation factor, $\varepsilon$ is the total fractionation factor(SI Appendix, Table S1).

In the three-component mixing model, the isotope of precipitation water vapor from the upwind direction should be known.In this study, it is determined that the advected water vapor ($\delta_{adv}$) in the oasis area comes from the desert region (Zhu et al., 2019),which can be calculated using the Rayleigh distillation equation:

$$\delta_{adv} = \delta_{pv-adv} + (\alpha^+ - 1)lnF, \tag{10}$$

Where $\delta_{pv-adv}$ is the isotope composition of precipitation water vapor from the upwind direction, F is the ratio between the final vapor and the initial vapor. As it is difficult to obtain the ratio between final and initial vapor, it can also be obtained through the ratio of average land surface vapor pressure between the two regions.

### 3.3.3 Thermodynamic transport-based raindrop re-evaporation model

Based on the traditional raindrop re-evaporation model, the heat and mass transfer states during raindrop falling are reconsidered, and it is assumed that the stable isotopes of raindrops released at the lifting condensation level (LCL) are in equilibrium fractionation with the background water vapor, that is, the influence of cloud layers on the stable isotopes of precipitation during the rainfall period in the convection process is not considered (Xia et al., 2021). Then, when dt=0.1s during the raindrop falling process, the formula for calculating the change in raindrop mass (dm) is as follows:

$$\frac{dm}{dt} = \frac{2\pi d f_v D}{R_w}\left(h\frac{e_{sat.T_a}}{T_a} - \frac{e_{sat.T_d}}{T_d}\right), \tag{11}$$

Where d represents the raindrop diameter, $f_v$ is the mass transfer coefficient for water vapor(Pruppacher and Klett, 2010), D and Rw are the diffusion coefficient and specific gas constant for water vapor, respectively, h represents the relative humidity,





$e_{sat.Ta}$ and $e_{sat.Td}$ represent the saturated vapor pressure of the ambient air (at ambient temperature Ta) and the boundary layer in contact with the raindrop (at raindrop temperature Td), respectively.

The equations for calculating the latent heat L (Jkg-1) during raindrop evaporation and the thermal conductivity of air ka (Jm-1s-1K-1) are given as(Rogers and Yau, 1989):

$$L = 2500800 - 2360(T_d - 273.15) + 1.6(T_d - 273.15)^2 - 0.06(T_d - 273.15)^3, \tag{12}$$

$$k_a = 418.4[5.69 + 0.017(T_d - 273.15)]10^{-5}, \tag{13}$$

The raindrop temperature $T_d$ is initially equal to Ta when the raindrop forms, As it falls, the rate of change in $T_d$ balances with the latent heat loss due to evaporation and the sensible heat flux from the warm air in lower layers. The equation for this calculation is as follows:

$$\frac{dT_d}{dt} = \frac{12}{d^2 \rho_w c_w} \left[ \frac{L f_v D}{R_w} \left( h \frac{e_{sat.T_a}}{T_a} - \frac{e_{sat.T_d}}{T_d} \right) - f_h k_a (T_d - T_a) \right], \tag{14}$$

Where ρw and cw are the density and specific heat of liquid water, respectively(Salamalikis et al., 2016), L is the latent heat of evaporation. $f_h$ and ka are the thermal ventilation coefficient and thermal conductivity of air, respectively.

### 3.3.4 Microphysical Model Validation Parameters

A crucial initial condition in the raindrop re-evaporation model is to specify the diameter (d) of raindrops at the LCL for the model. To ensure the accuracy of the model's initial parameters, the final raindrop radius (r) calculated from the
microphysical model is used for back-calculation to determine the initial raindrop diameter dfinal. In the microphysical model, the final raindrop radius r is calculated based on daily rainfall. The residual percentage of raindrops is obtained by computing the ratio of final raindrop mass to evaporated raindrop mass, with the calculation formula given as follows(Wang et al., 2024):

$$r = 0.2374 + 0.7298(1 - e^{-0.1243P}), \tag{15}$$

Where r denotes the radius of the raindrop upon reaching the ground surface (mm), and P denotes the daily precipitation (mm).

$$m_{end} = \frac{4\rho\pi r^3}{3}, \tag{16}$$

$$m_{ev} = \frac{E \cdot LCL}{v_{end}}, \tag{17}$$

$$f_{mic} = \frac{m_{end}}{m_{end} + m_{ev}}, \tag{18}$$

Where mend denotes the mass of the raindrop upon reaching the ground surface (g), E denotes the evaporation rate of precipitation (g/s), and LCL denotes the lifting condensation level of the raindrop (m). fmic represents the final residual fraction of the raindrop.





## 4 Results

### 4.1 Assessment of Oasis Vegetation Evapotranspiration

The variation characteristics of stable isotopes in precipitation are more easily influenced by temperature indicators. Comparing hydrogen and oxygen isotope values of precipitation in different cover types in the oasis area, it was found that the stable isotopes in tree precipitation and soil water were more enriched, while those in xylem water were more depleted. Throughout the growing season, the average $\delta^2H$ value of forest precipitation was -35.15‰ (Table 1), and the average $\delta^{18}O$ value was -5.76‰. In summer, isotope enrichment was more pronounced, with average hydrogen and oxygen isotope values

of -23.38‰ and -3.98‰, respectively. The isotope fractionation of soil water in farmland was smaller and more depleted, with an average $\delta^2H$ value of -64.73‰ and an average $\delta^{18}O$ value of -8.83‰. Compared to deep soil water, the non-equilibrium evaporative fractionation of 0-10cm shallow soil water was more intense, with a large range of isotope values across months. The hydrogen and oxygen isotope values of shallow soil water were highest in July at -26.49‰ and 1.34‰, respectively, and lowest in June at -79.92‰ and -9.96‰, respectively. In the process of water transfer from precipitation to

soil to plants, transpiration gradually became one of the dominant factors influencing the isotope variation in xylem water. Therefore, compared to other water sources, the response relationship between xylem water isotopes and temperature and precipitation was not obvious. The evaporation of water in crop stems was stronger, with an average $\delta^2H$ value of -48.58‰ and an average $\delta^{18}O$ value of -4.39‰, followed by grassland, with hydrogen and oxygen isotope values of -49.13‰ and -2.20‰, respectively.

**Table 1: Variations in stable isotopes of precipitation, soil water, and xylem water in oasis vegetation.**

| Land cover type | Month | precipitation | | soil water | | xylem water | |
|---|---|---|---|---|---|---|---|
| | | $\delta^2H$/‰ | $\delta^{18}O$/‰ | $\delta^2H$/‰ | $\delta^{18}O$/‰ | $\delta^2H$/‰ | $\delta^{18}O$/‰ |
| | April | -61.27 | -8.89 | -62.38 | -7.77 | / | / |
| | May | -53.20 | -7.63 | -64.16 | -8.45 | -43.41 | -0.06 |
| | June | -38.33 | -5.25 | -64.56 | -8.53 | -49.14 | -3.73 |
| | July | -51.78 | -7.56 | -66.47 | -9.02 | -55.99 | -5.54 |
| cropland | August | -51.64 | -7.59 | -68.59 | -9.34 | -53.78 | -6.74 |
| | September | -43.86 | -6.73 | -59.87 | -8. 39 | -40.59 | -5.89 |
| | October | / | / | / | / | / | / |
| | April | -51.19 | -7.58 | -46.53 | -2.14 | -57.23 | -2.42 |
| | May | -49.25 | -7.66 | -44.61 | -5.54 | -56.26 | -6.55 |
| | June | -24.48 | -4.71 | -48.12 | -3.31 | -61.09 | -0.86 |



| | | | | | | | |
|---|---|---|---|---|---|---|---|
| tree | July | -22.18 | -3.12 | -43.27 | -1.58 | -47.25 | 0.43 |
| | August | -23.48 | -4.11 | -52.27 | -7.13 | -70.58 | -6.99 |
| | September | -28.39 | -5.73 | -56.38 | -8.24 | -76.09 | -7.44 |
| | October | -47.13 | -7.44 | -58.19 | -8.49 | -52.45 | -6.87 |
| | April | -85.47 | -12.63 | -57.59 | -7.61 | / | / |
| | May | -57.72 | -8.13 | / | / | -42.38 | -0.97 |
| | June | -34.41 | -5.48 | -48.66 | -3.14 | -41.76 | -1.16 |
| | July | -29.53 | -4.86 | -54.53 | -5.77 | -48.74 | -1.71 |
| grassland | August | -32.98 | -5.55 | -48.97 | -4.98 | -48.57 | -2.04 |
| | September | -34.01 | -6.46 | -47.61 | -5.57 | -64.21 | -5.14 |
| | October | -54.07 | -9.57 | -60.53 | -7.65 | / | / |
| | April | -67.42 | -9.95 | -43.01 | -1.27 | -39.47 | 4.49 |
| | May | -59.57 | -8.91 | -65.52 | -9.29 | -48.77 | -3.12 |
| | June | -40.48 | -5.88 | -47.22 | -4.28 | -40.48 | -5.88 |
| shrub | July | -31.54 | -3.36 | -44.87 | -3.58 | -37.37 | -2.17 |
| | August | -45.43 | -6.07 | -60.53 | -8.43 | -78.58 | -7.70 |
| | September | -30.49 | -5.47 | / | / | -76.09 | -7.44 |
| | October | -66.54 | -9.66 | / | / | / | / |

Vegetation transpiration is a more significant and complex component of ecosystem evapotranspiration. Based on a single $\delta^{18}O$ analysis of the isotopic composition of evapotranspiration for four vegetation types, results show that the $\delta_{ET}$ value of cropland was highest in summer at -11.39‰ (Figure 2, a), and compared to the other three vegetation types, crops had the highest transpiration ratio ($f_T$) at 62%. The $f_T$ of trees and shrubs were similar, both at 45%, while grassland had a lower $f_T$. In summer, the average vegetation transpiration ratio of cropland was 62.51%. Grassland had the lowest $\delta_{ET}$, indicating a lower evapotranspiration capacity, with a vegetation transpiration ratio of only 40.70%. On a temporal scale, in summer, due to higher temperatures, the oxygen isotope values of $\delta_{ET}$, $\delta_E$, and $\delta_T$ were all greater than in spring and autumn. On a spatial scale, cropland and grassland had higher $\delta_T$ values at -4.39‰ and -2.20‰ respectively (Figure 2, b), while trees had the highest $\delta_E$ value at -31.48‰. This indicates that transpiration in cropland and grassland led to more enriched isotopes in xylem water, while the isotopes of water vapor transpired into the air were relatively depleted. Trees had a higher degree of soil evaporation fractionation, and the water vapor evaporated from the soil was also relatively depleted.



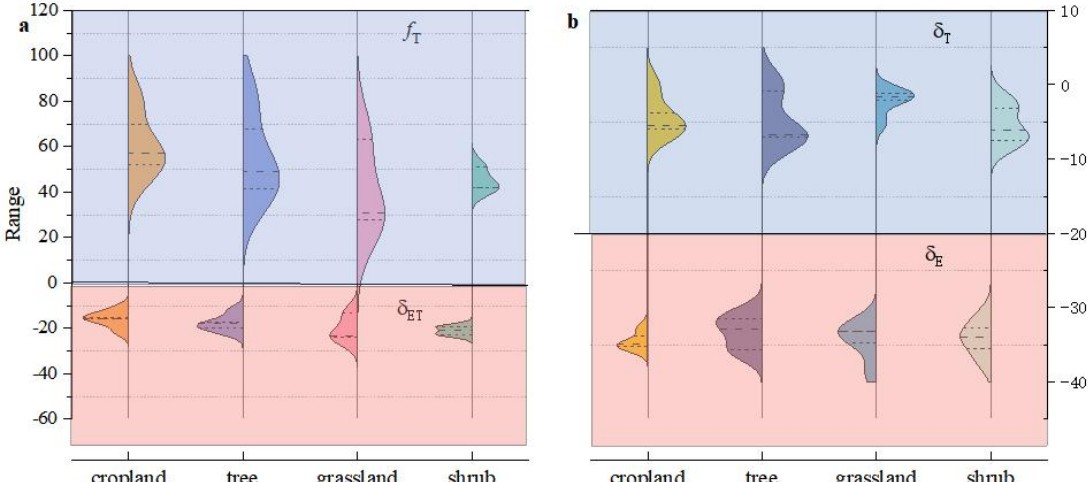

**Figure 2: Evapotranspiration capacity of different vegetation ecosystems. a represents the evapotranspiration composition ($\delta_{ET}$) and transpiration ratio ($f_T$) of the ecosystem, b represents the oxygen isotope composition ($\delta_E$, $\delta_T$) of soil evaporation and vegetation transpiration.**

## 4.2 Effects of evaporated water vapor on precipitation

### 4.2.1 Influence of surface evapotranspiration on precipitation

As temperatures rise to about 20°C in May and rainfall gradually increases, ground vegetation progressively accelerates water consumption through evapotranspiration and enhances its organ growth. From May to October, the intensity of stable isotope fractionation in water vapor peaks during summer. Compared to vertical water vapor, the advected water vapor is more enriched due to the shorter distance from the downwind origin, leading to higher average hydrogen and oxygen isotope values for $\delta_{adv}$ at -100.74‰ and -12.42‰ respectively (Figure 3, a). Rainfall water vapor around trees is higher than in other areas, particularly in July and August, with $\delta^2H$ peaking at -99.28‰ and $\delta^{18}O$ at -12.54‰, suggesting that precipitation in this area is more intensely affected by evaporation. Evaluation of the proportions of various water vapor sources in precipitation across different oasis regions reveals that the contribution of transpiration vapor from trees and croplands exceeds not only the evaporation vapor from surrounding soil but also that from grasslands and shrubs, at 27% and 25% respectively (Figure 3, b). This suggests that transpiration from trees and crops substantially elevates atmospheric water content. In the comprehensive analysis of recycled water vapor, trees exhibit the highest recycled water vapor ratio ($f_{re}$) at 53%, while shrubs show the lowest at only 30%.





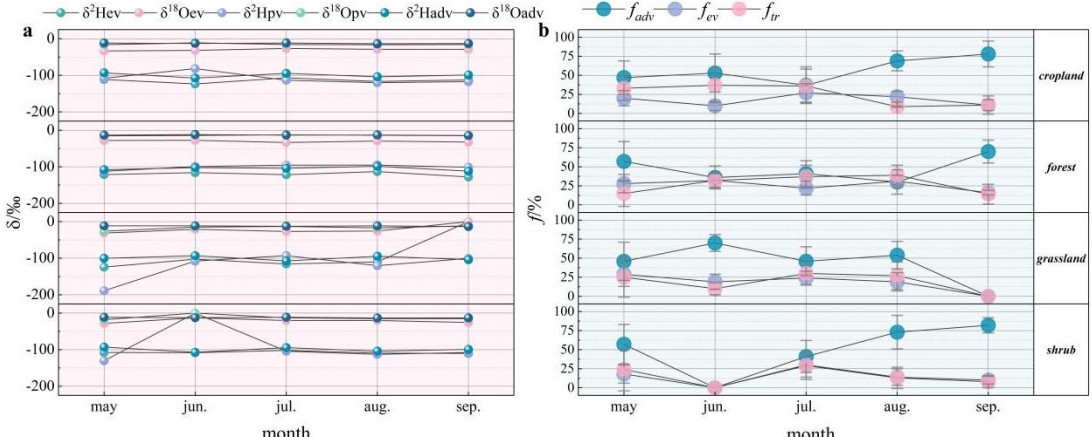

**Figure 3: Isotopic composition of recycled water vapor during the growing season (a) and its contribution ratio to precipitation (b).**

### 4.2.2 Mass lossof raindrop due to secondary evaporation sub-cloud

Near-surface temperature and relative humidity determine the lifting condensation level (LCL) and descent time of raindrops.
210 Temperature and LCL show roughly the same trend (Figure 4, a). Based on the hourly rainfall in the study area, it is determined that the initial diameter of raindrops falling from cloud base could be 0.5mm (drizzle), 1mm (normal rain), and 1.5mm (heavy rain). When the initial raindrop diameter (d) is 0.5mm, the re-evaporation rate of raindrops for all vegetation types is essentially 100% (Figure 4, b), with the raindrop diameter being nearly 0 upon reaching the ground. As rainfall intensity gradually increases and d changes to 1mm, the average re-evaporation rate of raindrops above croplands, grasslands,
215 and shrubs is between 60-70%, while for trees it is only 39%. The change in $\delta^{18}O$ during raindrop fall is affected by rainfall amount. As rainfall increases, the change value slowly increases from 1‰ (Figure 4, c). The diameter of raindrops when reaching the ground is not only related to LCL but also regulated by the re-evaporation rate. Regardless of whether it's heavy rain, moderate rain, or light rain, trees consistently have the largest final raindrop diameter, which is a result of lower LCL and smaller re-evaporation rate.



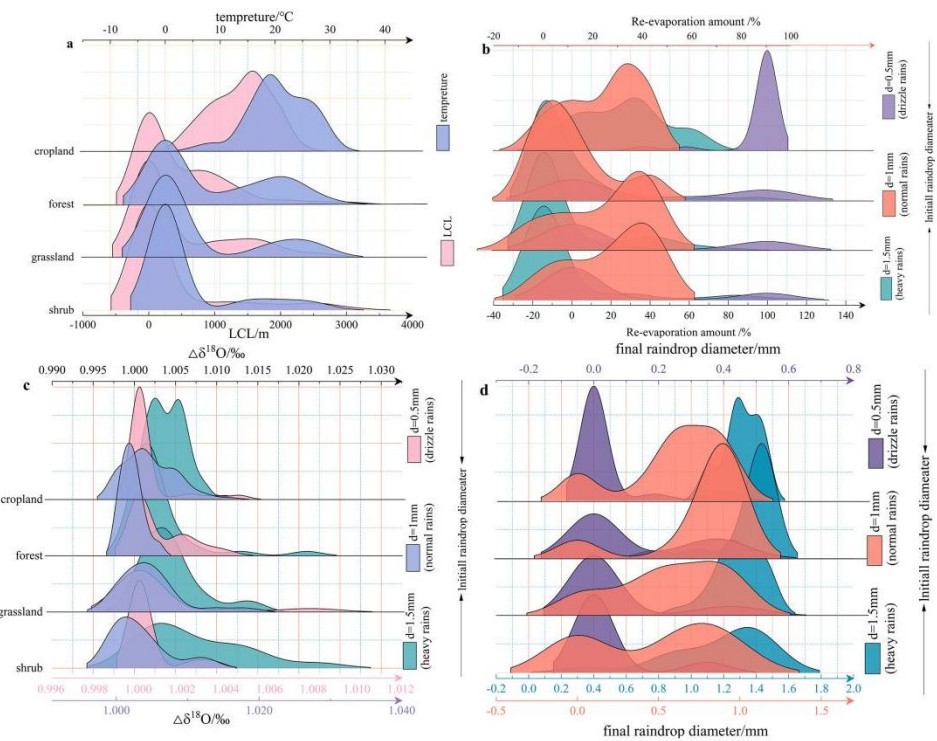

**Figure 4: Sub-cloud secondary evaporation loss of raindrops. a represents the lifting condensation level of raindrops, b represents the re-evaporation rate of raindrops, c represents the change in $\delta^{18}O$ value during raindrop fall, d represents the diameter of raindrops after evaporation.**

## 4 Discussion

### 5.1 Precipitation Changes Caused by Evapotranspiration

In arid regions, atmospheric water vapor can reach saturated or unsaturated states through vegetation evapotranspiration. Although how vegetation and soil regulate diurnal temperature range in arid regions remains unknown (Zhou et al., 2007), precipitation is highly sensitive to changes in plant diversity at the local scale (Korell et al., 2021). Analysis shows that except for grasslands, the other three vegetation types have a more pronounced correlation between rainfall and $f_{re-ev}$ (Figure 5, a,b,d), while grassland rainfall is mainly influenced by recycled water vapor (Figure 5, b). This indicates that both vegetation evapotranspiration and sub-cloud secondary evaporation of raindrops help accelerate local water vapor circulation, increasing the rate of precipitation formation under suitable temperature conditions. The low correlation between daily soil evaporation and daily precipitation in croplands (Figure 5, e) indicates that in the vertical direction of water vapor transport, vegetation transpiration is the main factor affecting atmospheric water vapor content and precipitation. Desert areas have an average daily rainfall of only 4mm, with an average raindrop re-evaporation rate as high as 80% throughout the growing





season. However, the harsh vegetation growth conditions in desert regions result in a recycled water vapor ratio of only about 15% (Zhu et al., 2019), implying that the underlying vegetation cannot enhance rainfall.

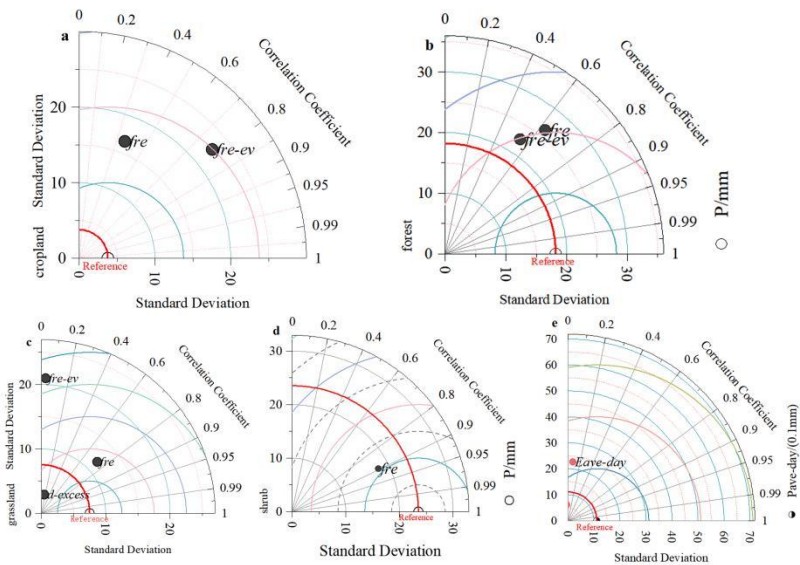

**Figure 5: Correlation between rainfall and evaporated water vapor for different vegetation types. a, b, c, d represent the**
**correlation between monthly rainfall during the growing season and recycled water vapor ($f_{re}$) and re-evaporated water vapor ($f_{re-ev}$), respectively, e represents the correlation between daily rainfall and daily evaporation in croplands.**

## 5.2 Regulation of Water Vapor Saturation by Vegetation and Atmosphere

Over the past 30 years, vegetation transpiration has accounted for up to 70% of the mitigation of global land surface temperature rise (Zeng et al., 2017). Evapotranspiration is central to water, energy, and carbon cycles, with a clear responsive relationship between evapotranspiration and vegetation greening at local/regional scales (Yang et al., 2023). Furthermore, vegetation alters atmospheric water vapor content by regulating ET changes and modifies lower atmosphere stability by redistributing surface energy, ultimately affecting precipitation. Enhanced precipitation recycling and water vapor convergence will significantly offset water vapor transport under monsoon and westerly influences, with the main driving force coming from gradually increasing evapotranspiration levels of surface vegetation. The amount of sub-cloud evaporation loss of raindrops is mainly determined by relative humidity, atmospheric temperature, and rainfall amount. Although croplands and forests have significantly higher average LCL values (Figure 6), grasslands and shrubs have higher relative humidity. Therefore, more water vapor from raindrop evaporation converges in the atmosphere above grasslands and shrubs. To ensure heat and moisture balance in the lower atmosphere, there is a clear negative feedback between vegetation



and atmosphere, meaning that simultaneous increases in vegetation evapotranspiration and raindrop re-evaporation ratios

will not occur in the atmosphere. As evapotranspiration increases, vertical water vapor recycling gradually accelerates, with

evaporative cooling effects driving changes in precipitation and temperature at the microclimate scale.

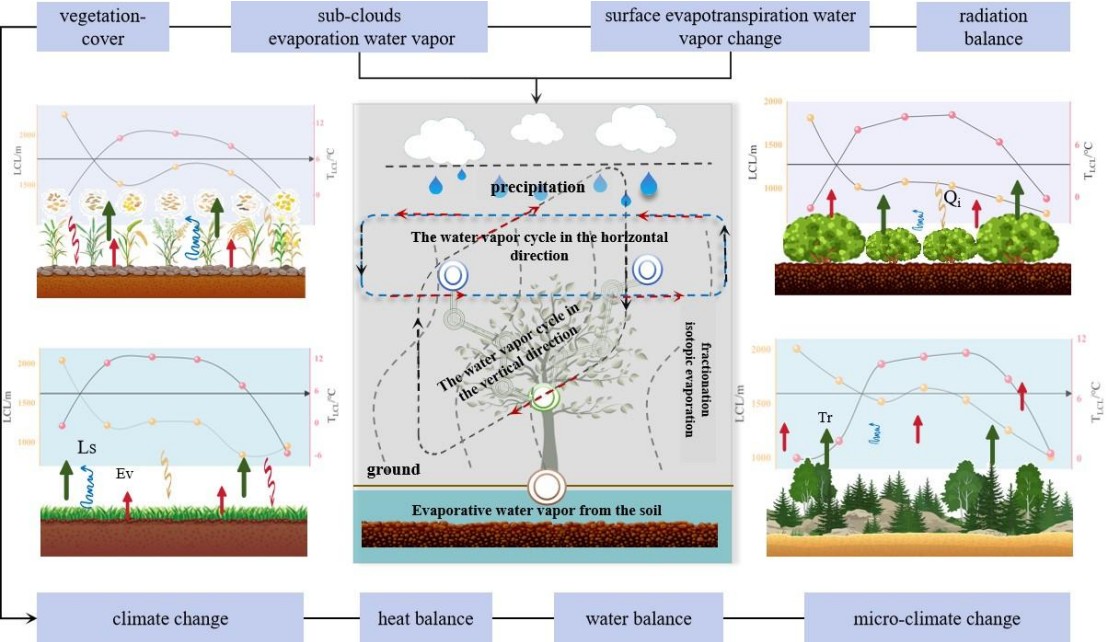

**Figure 6: The negative feedback mechanism for water vapor evaporation between vegetation and the atmosphere.**

### 5.3 Uncertainty analysis of the impact of oasis vegetation evapotranspiration on precipitation

Combining the daily rainfall in the study area, the final diameter of raindrops after falling was determined through a

microphysical model, ranging between 0.23-0.96mm (Figure 7, a). Based on the re-evaporation model, assuming an initial

raindrop diameter of 0.5mm (less than 1mm/h), the $d_{final}$ range is between 0 and 0.35mm, with an average of 0.03mm. For an

initial diameter of 1mm (less than 3mm/h), the $d_{final}$ range is between 0 and 0.94mm, with an average of 0.63mm. For an

initial diameter of 1.5mm (less than 15mm/h), the $d_{final}$ range is between 1.09 and 1.46mm, while an initial diameter increase

to 2.6mm (less than 30mm/h) is not applicable to this study area (Figure 7, b) (Xia et al., 2022). Therefore, based on the $d_{final}$

determined by the microphysical model, it is inferred that the possible initial diameter d of raindrops is between 0.5-1.5mm.

The two models calculate the remaining raindrop ratio ($f_{mic}$) and re-evaporated raindrop ratio ($f_{re-ev}$) with opposite trends, but

in terms of ratios, there is no significant relationship between the results of the two models (Figure 7, c,d).



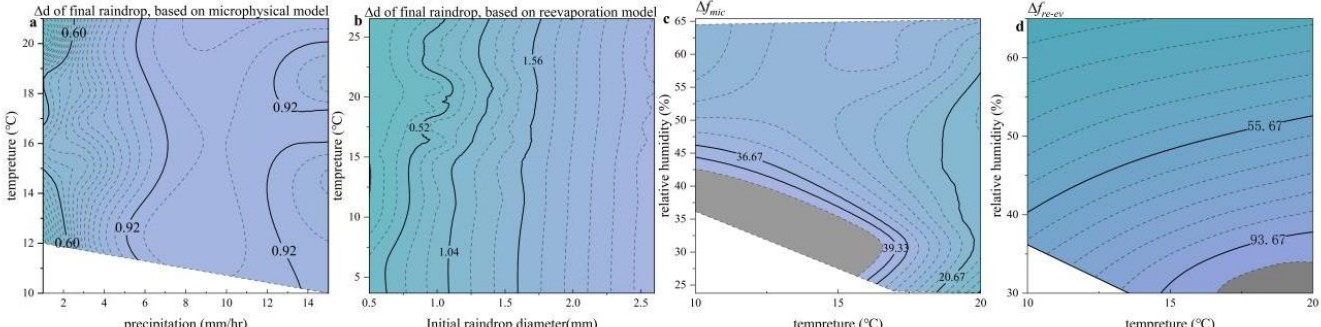

**Figure 7: Calibration of parameters and results in different models. a represents the possible changes in the final diameter of raindrops based on the microphysical model, b represents the initial and final diameters of raindrops based on the re-evaporation model, c represents the change in the remaining raindrop ratio ($f_{mic}$) after re-evaporation, d represents the change in the raindrop re-evaporation ratio ($f_{re-ev}$).**

## 6 Discussion

This study, using stable isotope analysis, found that the water consumption capacity of crop transpiration far exceeds soil evaporation, with an average contribution rate ($f_T$) of up to 62% in the entire ecosystem. Trees significantly increased atmospheric water content through recycled water vapor ($f_{re}$), with an average of 53%, while shrubs had the lowest at only 30%. When hourly rainfall is between 1 and 3mm, the average re-evaporation rate ($f_{re-ev}$) of raindrops above farmland, grassland, and shrubs is between 60-70%, while for trees it is only 39%. There is a clear negative feedback relationship

between precipitation re-evaporation vapor and vegetation evapotranspiration vapor in the atmosphere, indicating that vegetation evapotranspiration, especially after enhanced transpiration capacity, may inhibit or even reduce secondary evaporation loss of raindrops below clouds. The upward heat and energy balance from the surface regulates atmospheric moisture deficit, and by accelerating precipitation formation and local evaporative cooling effects, it modifies the microclimate environment, ultimately stabilizing the lower atmosphere. Our findings suggest that the diversity of underlying

surface vegetation is fundamental to maintaining ecosystem stability, while the circulation amount and rate of evapotranspiration vapor are crucial in determining atmospheric precipitation conditions and vegetation productivity.



**Data Availability**

The data that support the findings of this study are openly available at https://doi.org/10.17632/vhm44t74sy.1.

**Author contribution**

Yinying Jiao and Guofeng Zhu conceived the idea of the study; Yuxin Miao, Yani Gun, Jiangwei Yang participated in the drawing; Yinying Jiao wrote the paper; Qinqin Wang checked the format. All authors discussed the results and revised the manuscript.

**Competing interests**

The authors declare that they have no conflict of interest.

**Acknowledgments**

This research was financially supported by the National Natural Science Foundation of China (41971036, 42371040). Daily precipitation and evaporation dataset is provided by National Ecosystem Science Data Center, National Science & 300 Technology Infrastructure of China( http://www.nesdc.org.cn).

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
