# Peer review of "Enhanced Vegetation Evapotranspiration Increases Precipitation in Oasis Regions"

_EGUsphere, 2025_

## Author Comment (AC1)

**Responses to the reviewer's comments**

**Response to Reviewer #1**

First of all, we are truly grateful for your feedback. It means a great deal to us that you are willing to review and support our research. We genuinely appreciate the time and effort you have dedicated to evaluating our work.

We have carefully considered each of your comments and fully understand your concerns about the quality of the previous version. Your insights, along with those from another reviewer, have guided us to make significant improvements to this manuscript.

We believe that research based on field observations should be robust and withstand thorough scrutiny. We are committed to addressing all of your questions with sincerity and diligence. We respectfully ask for another opportunity to present this study, and we eagerly look forward to your continued guidance.

The red words, sentences, and subsections represent our editing changes.

**Responses to the general comments**

**Question 1**

(The manuscript is very weakly written. The structure is unclear and the English is poor. This results in sometimes incorrect statements, which is a major concern. Examples include calling plants "buffer zone" or stating that the growth of vegetation results from "the synergistic stimulation from increased water and nutrient supply under global warming.")

**Response:** Thank you very much for your valuable feedback. We truly appreciate your insights, and we have made significant revisions in response to your comments in two key areas.

**First**, we understand your concerns regarding the manuscript's language. We recognize how language issues can affect the overall perception of our work. To address this, we have carefully revised the entire manuscript. Since we're currently in a position to respond only to your comments, we'd like to share some portions of the revised abstract and introduction for your review.

**Second**, we acknowledge that there were inaccuracies in the abstract, introduction, and results sections. To correct these errors, we took the time to identify the sources of the problematic statements and carefully reviewed the relevant literature. We have removed any sentences that lacked clarity and have reorganized the references to ensure accuracy.

Below, we present the modifications we made to the abstract and introduction:

"Abstract. While the impact of vegetation on global climate has been confirmed, the feedback mechanisms between vegetation and precipitation at local scales remain unknown. This study selects oasis as relatively independent geographical units and analyzes stable isotopes in precipitation, soil water, and xylem water across four vegetation cover types. Results show that in oasis areas, tree-covered regions have the highest recycled vapor ratio (fre), averaging 53%, and the lowest raindrop re-evaporation rate (fre-ev) at 38%. Cropland, grassland, and shrub-covered areas have lower fre (39%)

and higher average fre-ev (60 - 70%). In addition, sub-cloud re-evaporation loss of raindrops lead to higher relative humidity around grasslands and shrubs. This implies that vegetation suppresses precipitation evaporation losses through transpiration, thereby maintaining water vapor balance in the lower atmosphere. This study provides new insights into how local vegetation influences precipitation changes, and suggests that the potential effects of large-scale ecosystem restoration in arid regions on water resource availability warrant re-examination.

**1 Introduction**

Globally, species that are better adapted to warmer environments are gaining an advantage. In recent decades, many temperate forests have experienced a continuous expansion of greening areas and a gradual increase in tree biomass, resulting in a canopy closure effect that buffers the impact of atmospheric warming on understory vegetation (De Frenne et al., 2013). Since the last glacial period, numerous terrestrial sediment records have shown evidence of rapid climate change (Peteet, 2000), including severe extreme climate events that have left irreversible traces on Earth's ecosystems. To mitigate the effects of climate change, plants have increased their productivity and ecosystem stability, and enhanced the stability of temperature and precipitation by acting as a buffer zone (Huang et al., 2024). Evaporation and cooling are key factors driven by vegetation that link surface energy redistribution with terrestrial water and carbon cycling processes (Wang and Zeng, 2024). In terms of the relationship between vegetation and the water cycle, as well as various hydrological elements, climate warming may reduce the amount of available surface water by promoting vegetation growth and enhancing ecosystem evapotranspiration. Recent studies have indicated that plants have increased their productivity and ecosystem stability, thereby stabilizing temperature and precipitation by acting as a buffer zone (Huang et al., 2023). Stable isotopes of hydrogen ( $\delta^2$ H) and oxygen ( $\delta^{18}$ O) in water bodies undergo fractionation during different processes in the water cycle, leading to changes in their isotopic compositions (Dansgaard et al., 1964). This implies that the stable isotopic composition varies among different water bodies, making hydrogen and oxygen valuable tracers for investigating the sources of water and the movement pathways of evaporated water vapor (Walker and Brunel, 1990; Gibson and Edwards, 2002). Oasis, characterized by a stable water supply and vegetation cover, form distinct geographical units within arid desert environments. Given the abundant precipitation in oasis areas and the ambiguous relationship between different vegetation types and precipitation, isotopes of hydrogen and oxygen present in precipitation, plant xylem water, and soil water serve as valuable tools for

elucidating the relationship between functional changes in ecosystems and climate variability, in accordance with the principles of water balance and isotopic mass balance.

Prolonged water scarcity can determine the sensitivity of biological communities to drought, leading to rapid vegetation responses to adapt to changing water resource mechanisms in short-term drought scenarios (Li et al., 2023). Transpiration from vegetation is a major component of terrestrial evapotranspiration (ET), serving as a crucial link between the water and carbon cycles at the land surface and the atmosphere. Given the dominant role of transpiration in land evapotranspiration, it possesses the capacity to influence regional precipitation and surface temperature by altering latent heat flux (Jasechko et al., 2013). Existing research has demonstrated that the recirculated water vapor formed by surface evapotranspiration can generate rainfall as a key source of moisture transport, thereby bridging the atmospheric moisture gap and alleviating regional drought conditions (Gimeno-Sotelo et al., 2024). Changes in evapotranspiration may also impact cloud formation and subsequent rainfall events (Zhang et al., 2023). During the process of raindrops descending from clouds to the ground, continuous exchange of water molecules occurs with the ambient water vapor, resulting in partial or complete evaporative losses in unsaturated air. This water vapor exchange process, primarily driven by secondary evaporation beneath the clouds, is closely related to the lifting condensation level (LCL) of the raindrops and directly affects the amount of precipitation reaching the surface. Furthermore, it influences boundary layer temperature and relative humidity through evaporative cooling (Graf et al., 2019). This also implies that changes in the near-surface thermodynamic state (temperature and humidity) induced by vegetation regulate the lifting condensation level (LCL) and the re-evaporation rate of raindrops, potentially creating a feedback loop that affects precipitation intensity and isotopic characteristics. Currently, research on this process remains relatively unclear, particularly regarding the influence of different vegetation types on rainfall.

As a relatively humid enclave at the desert margin, the oasis's special geographical location not only has important ecological service value but also greatly promotes the social and economic development of the region. Oasis vegetation has typical drought-tolerant and salt-resistant abilities, which can be mainly divided into three categories: trees, shrubs, and grasses. Moreover, the relatively abundant water resources provide favorable conditions for the development of irrigated agriculture in the region. In this study, we use stable isotopes of water to determine the sensitivity of precipitation to changes in evapotranspiration of different vegetation types and to address the following issues: 1) Quantifying the evapotranspiration ratios of forest, farmland, grassland, and shrub ecosystems in arid oasis areas; 2) Assessing the impact of local water vapor recycling on precipitation. This study focuses on the

contribution of vegetation evapotranspiration to precipitation and examines how water vapor is transported. By looking into the relationship between vegetation evapotranspiration and precipitation, the results are helpful for identifying the possible reasons for changes in water availability in oasis areas."

**Question 2**

(The introduction of the research aim is too weak. Apparently the authors aim at "solving the problem of precipitation distribution patterns", but what problem this is exactly is unclear. The isotopes appear out of nowhere, and introduction section does not include any introduction of the **study area**, leaving the reader guessing what is meant by "the oasis" or "the region". This is illustrative of the unjustifiably generic claims that the manuscript makes, leading to overpromises of the results. It is sometimes hard to judge the merits of the work, because of blurriness between findings and interpretations (example: lines 169-172), but surely generic claims such as in the title and abstract, but also in the following sentences, are not justified:

"The research results will help fully elucidate the impact of vegetation evapotranspiration on precipitation and consider the regulatory effect of vegetation on precipitation and atmospheric water vapor from the perspective of heat transfer."

"This study, using stable isotope analysis, found that the water consumption capacity of crop transpiration far exceeds soil evaporation.")

**Response:** Your concerns are crucial, and we will take each of your questions and suggestions very seriously.

1. In the abstract, we have added an explanation of the principles of isotopes in water bodies and introduced the concept of oasis areas. The details are as follows:

Stable isotopes of hydrogen ( $\delta^2$ H) and oxygen ( $\delta^{18}$ O) in water bodies undergo fractionation during different processes in the water cycle, leading to changes in their isotopic compositions (Dansgaard et al., 1964). This implies that the stable isotopic composition varies among different water bodies, making hydrogen and oxygen valuable tracers for investigating the sources of water and the movement pathways of evaporated water vapor (Walker and Brunel, 1990; Gibson and Edwards, 2002). Oasis, characterized by a stable water supply and vegetation cover, form distinct geographical units within arid desert environments. Given the abundant precipitation in oasis areas and the ambiguous relationship between different vegetation types and precipitation, isotopes of hydrogen and oxygen present in precipitation, plant xylem water, and soil water serve as valuable tools for elucidating the relationship between functional changes in ecosystems and climate variability, in accordance with the principles of water balance and isotopic mass balance.

2. We have removed vague and unreasonable sentences from the abstract, introduction, and results

sections. For example: "The research results will help fully elucidate the impact of vegetation evapotranspiration on precipitation and consider the regulatory effect of vegetation on precipitation and atmospheric water vapor from the perspective of heat transfer." "This study, using stable isotope analysis, found that the water consumption capacity of crop transpiration far exceeds soil evaporation."

3.We are genuinely concerned about whether the title is appropriate and how it might be improved. We find ourselves at a bit of a loss in this regard, so we would like to humbly and sincerely seek your advice. We are committed to making the necessary modifications to the title to enhance its relevance and clarity.

**Question 3**

Beyond the quality of writing, sloppy mistakes in the text and figures make it difficult to trust the carefulness of the work. The authors need to carefully consider what to show and how, and take the reader along in the message, which is not the case now. For example, three new figures are presented in the Discussion, including a "calibration of different models" as the final figure. Section 5 is headed "4: Discussion", but also section 6 is headed "Discussion".

**Response:** We feel sincerely embarrassed about the issues and errors present in our manuscript. Nevertheless, we genuinely hope you might consider a second review of our work and give it another chance. We are committed to thoroughly checking for all errors in the manuscript and actively correcting them.

We sincerely request the reviewer #1 to re-evaluate the revised manuscript. Once again, thank you for your valuable feedback; we will ensure that the changes significantly enhance the clarity, credibility, and scientific quality of the paper.

---

## Author Comment (AC2)

**Responses to the reviewer's comments**

**Response to Reviewer #2**

We sincerely appreciate Reviewer #2's recognition of the scientific merits of our manuscript and the valuable suggestions provided. We are confident that, following these major revisions, the quality of our manuscript will be substantially improved.

We will address every one of your comments with the utmost care, and we hope that, after our revisions, the manuscript will meet your expectations and justify the time and effort you have generously invested. The red words, sentences, and subsections represent our editing changes.

**Responses to the general comments**

"However, the manuscript in its current form requires major revision to meet the publication standards of HESS. The central conclusion regarding plant transpiration increasing precipitation requires more rigorous substantiation. The logical relationships and computational methodologies underpinning this conclusion must be carefully scrutinized and strengthened. Furthermore, a critical oversight is the lack of discussion regarding the role of groundwater irrigation in sustaining the oasis ecosystem, which is fundamental to the hydrological cycle described."

**Response:** Thank you very much for your affirmation and support of our research work. We have carefully revised each of your suggestions and responded to them one by one. As a result of these revisions, the quality of the manuscript will be significantly improved.

Since irrigation water is primarily collected from local underground wells and serves as a critical water source for replenishing the soil and supporting vegetation in the study area. Therefore, we have included irrigation water data collected in 2019 and 2021. We have also added a discussion on the impact of irrigation water on evapotranspiration and recirculation in farmland.

"While the overall narrative is readable, the logical framework requires consolidation, and the results section presents findings that are currently vague and require clearer articulation. I strongly recommend a thorough revision of the language to eliminate ambiguity, particularly in key sentences conveying the core findings. I encourage the authors to address all points raised below proactively."

**Response:**We have checked for potential language issues in the manuscript and improved the readability of the abstract, introduction, results, and discussion sections to ensure that the main findings are presented more clearly to the readers.

**Responses to the major comments**

**Comment one:**

"The abstract does not clearly articulate the inhibitory relationship between recycled water vapor (fre) and re-evaporation vapor (fre-ev). The findings of Jiao et al. should be contextualized: in regions with high recycled vapor, sub-cloud secondary evaporation loss is minimal. The vegetation-precipitation

response narrative should also integrate key meteorological variables such as near-surface air temperature and relative humidity."

**Response:** Thank you for your valuable suggestions. We fully agree with your point that the intensity of raindrop evaporation losses under clouds is primarily determined by temperature, relative humidity, and precipitation. The discussion in this section mainly focuses on Section 5.2: Regulation of Water Vapor Saturation by Vegetation and Atmosphere, and the content is as follows:

"The amount of sub-cloud evaporation loss of raindrops is mainly determined by relative humidity, atmospheric temperature, and rainfall amount. Although croplands and forests have significantly higher average LCL values (Figure 6), grasslands and shrubs have higher relative humidity. Therefore, more water vapor from raindrop evaporation converges in the atmosphere above grasslands and shrubs. To ensure heat and moisture balance in the lower atmosphere, there is a clear negative feedback between vegetation and atmosphere, meaning that simultaneous increases in vegetation evapotranspiration and raindrop re-evaporation ratios will not occur in the atmosphere. As evapotranspiration increases, vertical water vapor recycling gradually accelerates, with evaporative cooling effects driving changes in precipitation and temperature at the microclimate scale."

We have summarized this part into the abstract. Our modifications are as follows:

"In addition, sub-cloud re-evaporation loss of raindrops lead to higher relative humidity around grasslands and shrubs. This implies that vegetation suppresses precipitation evaporation losses through transpiration, thereby maintaining water vapor balance in the lower atmosphere."

**Comment two:**

"The concluding statement on water resource assessments should be reframed to focus on the study's specific implications. Replace the current sentence with: "This finding also implies that the potential impacts of large-scale ecosystem restoration in arid regions on water resource availability must be re-examined."

**Response:** We have revised this sentence according to your suggestions. The details are as follows:

"This study provides new insights into how local vegetation influences precipitation changes, and suggests that the potential effects of large-scale ecosystem restoration in arid regions on water resource availability warrant re-examination."

**Comment three:**

"The introduction requires significant improvement in logical coherence, readability, and reference accuracy.

Logical Reorganization: The flow of argumentation should be restructured as follows:

- a) Global context and research motivation.
- b) The process mechanisms (e.g.,  $ET \rightarrow moisture\ budget \rightarrow boundary\ layer/LCL \rightarrow sub-cloud\ re-evaporation \rightarrow precipitation)\ that form the basis of the scientific questions.$

- c) Identification of the research gap and a clear statement of the research hypotheses.
- *d) Justification of the study area's typicality and particularity.*
- e) Clear articulation of the research objectives and significance.

Explicit Hypotheses: The following two hypotheses need to be explicitly stated:

- a) Different vegetation types alter the proportion of recycled water vapor by modifying evapotranspiration (ET) fluxes and the partitioning of its components.
- b) Vegetation-induced changes in the near-surface thermodynamic state (temperature and humidity) modulate the lifting condensation level (LCL) and raindrop re-evaporation rate, creating a feedback loop to precipitation intensity and isotopic signatures.

Reference Accuracy: Correct "De et al., 2013" to "De Frenne et al., 2013".

Focus: The second paragraph is tangential to the paper's focus on hydrological processes. Its content should be refined to progressively narrow the scope toward water-vapor feedback processes."

**Response:** We have improved the introduction in terms of logical coherence, readability, and reference accuracy.

First, we have reorganized the logic of the introduction according to your suggestions, especially in the second paragraph, which has undergone significant revisions. Additionally, we have clarified the two hypotheses of the study in the second paragraph and carefully checked the accuracy of the references in the introduction. The details of the modifications are as follows:

**"1 Introduction**

Globally, species that are better adapted to warmer environments are gaining an advantage. In recent decades, many temperate forests have experienced a continuous expansion of greening areas and a gradual increase in tree biomass, resulting in a canopy closure effect that buffers the impact of atmospheric warming on understory vegetation (De Frenne et al., 2013). Since the last glacial period, numerous terrestrial sediment records have shown evidence of rapid climate change (Peteet, 2000), including severe extreme climate events that have left irreversible traces on Earth's ecosystems. To mitigate the effects of climate change, plants have increased their productivity and ecosystem stability, and enhanced the stability of temperature and precipitation by acting as a buffer zone (Huang et al., 2024). Evaporation and cooling are key factors driven by vegetation that link surface energy redistribution with terrestrial water and carbon cycling processes (Wang and Zeng, 2024). In terms of the relationship between vegetation and the water cycle, as well as various hydrological elements, climate warming may reduce the amount of available surface water by promoting vegetation growth and enhancing ecosystem evapotranspiration. Recent studies have indicated that plants have increased their productivity and ecosystem stability, thereby stabilizing temperature and precipitation by acting as a

buffer zone (Huang et al., 2023). Stable isotopes of hydrogen ( $\delta^2H$ ) and oxygen ( $\delta^{18}O$ ) in water bodies undergo fractionation during different processes in the water cycle, leading to changes in their isotopic compositions (Dansgaard et al., 1964). This implies that the stable isotopic composition varies among different water bodies, making hydrogen and oxygen valuable tracers for investigating the sources of water and the movement pathways of evaporated water vapor (Walker and Brunel, 1990; Gibson and Edwards, 2002). Oasis, characterized by a stable water supply and vegetation cover, form distinct geographical units within arid desert environments. Given the abundant precipitation in oasis areas and the ambiguous relationship between different vegetation types and precipitation, isotopes of hydrogen and oxygen present in precipitation, plant xylem water, and soil water serve as valuable tools for elucidating the relationship between functional changes in ecosystems and climate variability, in accordance with the principles of water balance and isotopic mass balance.

Prolonged water scarcity can determine the sensitivity of biological communities to drought, leading to rapid vegetation responses to adapt to changing water resource mechanisms in short-term drought scenarios (Li et al., 2023). Transpiration from vegetation is a major component of terrestrial evapotranspiration (ET), serving as a crucial link between the water and carbon cycles at the land surface and the atmosphere. Given the dominant role of transpiration in land evapotranspiration, it possesses the capacity to influence regional precipitation and surface temperature by altering latent heat flux (Jasechko et al., 2013). Existing research has demonstrated that the recirculated water vapor formed by surface evapotranspiration can generate rainfall as a key source of moisture transport, thereby bridging the atmospheric moisture gap and alleviating regional drought conditions (Gimeno-Sotelo et al., 2024). Changes in evapotranspiration may also impact cloud formation and subsequent rainfall events (Zhang et al., 2023). During the process of raindrops descending from clouds to the ground, continuous exchange of water molecules occurs with the ambient water vapor, resulting in partial or complete evaporative losses in unsaturated air. This water vapor exchange process, primarily driven by secondary evaporation beneath the clouds, is closely related to the lifting condensation level (LCL) of the raindrops and directly affects the amount of precipitation reaching the surface. Furthermore, it influences boundary layer temperature and relative humidity through evaporative cooling (Graf et al., 2019). This also implies that changes in the near-surface thermodynamic state (temperature and humidity) induced by vegetation regulate the lifting condensation level (LCL) and the re-evaporation rate of raindrops, potentially creating a feedback loop that affects precipitation intensity and isotopic characteristics. Currently, research on this process remains relatively unclear, particularly regarding the influence of different vegetation types on rainfall.

As a relatively humid enclave at the desert margin, the oasis's special geographical location not only has important ecological service value but also greatly promotes the social and economic development of the region. Oasis vegetation has typical drought-tolerant and salt-resistant abilities, which can be mainly divided into three categories: trees, shrubs, and grasses. Moreover, the relatively abundant water resources provide favorable conditions for the development of irrigated agriculture in the region. In this study, we use stable isotopes of water to determine the sensitivity of precipitation to changes in evapotranspiration of different vegetation types and to address the following issues: 1) Quantifying the evapotranspiration ratios of forest, farmland, grassland, and shrub ecosystems in arid oasis areas; 2) Assessing the impact of local water vapor recycling on precipitation. This study focuses on the contribution of vegetation evapotranspiration to precipitation and examines how water vapor is transported. By looking into the relationship between vegetation evapotranspiration and precipitation, the results are helpful for identifying the possible reasons for changes in water availability in oasis areas."

**References**

Jasechko S, Sharp Z D, Gibson J J, et al. Terrestrial water fluxes dominated by transpiration[J]. Nature, 2013, 496(7445): 347-350.

Gimeno-Sotelo L, Sorí R, Nieto R, et al. Unravelling the origin of the atmospheric moisture deficit that leads to droughts[J]. Nature Water, 2024, 2(3): 242-253.

Graf P, Wernli H, Pfahl S, et al. A new interpretative framework for below-cloud effects on stable water isotopes in vapour and rain[J]. Atmospheric chemistry and physics, 2019, 19(2): 747-765.

**Comment four:**

**"Study Area:**

Clarify whether the value of 2580.7 mm refers to actual evapotranspiration or potential evapotranspiration."

**Response:** By reviewing the literature, we can confirm that 2580.7 mm refers to the potential evapotranspiration. To ensure the accuracy of the precipitation and potential evapotranspiration, we made the following modifications:

"The annual precipitation in the basin ranges from 54 to 608 mm, while the annual potential evaporation varies between 2000 and 3000 mm (Wang et al., 2012; Sang et al., 2023)."

**References**

Wang Z, Ficklin D L, Zhang Y, et al. Impact of climate change on streamflow in the arid Shiyang River Basin of northwest China[J]. Hydrological Processes, 2012, 26(18): 2733-2744.

Zongxing L, Qi F, Wang Q J, et al. Contributions of local terrestrial evaporation and transpiration to

precipitation using  $\delta$ 18O and D-excess as a proxy in Shiyang inland river basin in China[J]. Global and Planetary Change, 2016, 146: 140-151.

Sang L, Zhu G, Xu Y, et al. Effects of agricultural large-and medium-sized reservoirs on hydrologic processes in the arid Shiyang River Basin, Northwest China[J]. Water Resources Research, 2023, 59(2): e2022WR033519.

**"Materials and Methods:**

Sampling Details: The sampling methodology is insufficiently detailed. Please provide specific geographic coordinates of the sampling sites, the number of samples collected, and the plant species sampled.

Equation Notation: Carefully check the consistency of all subscripts and symbols in the equations, both in the main text and in the Supporting Information. Ensure all equations are appropriately referenced. Supporting Information: Place the citation for Table S1 in a more prominent location within the methodology section."

**Response:** We have supplemented the geographic locations of the sampling points, as well as the information on vegetation samples and sample quantities. In addition, we have reorganized the content of Section 3.1 Sample Collection. The details of the modifications are as follows:

**"3.1 Sample collection**

From April 2018 to October 2022, we established four sampling points in the middle and lower reaches of the Shiyang River basin (Table 1) to collect precipitation, irrigation water, soil water, and vegetation xylem water. For samples other than precipitation, the collection frequency was once a month during the growing season.

Table 1: Geographic Locations of Sampling Points, Sample Types, and Quantities

| Number | Elevation (m) | Geographic Location | Sample Type                                                                                                     | Sample Quantity (pcs) |
|--------|---------------|---------------------|-----------------------------------------------------------------------------------------------------------------|-----------------------|
| S1     | 1349          | 103°14'E, 38°46'N   | Corn stalk, 0-100 cm soil, irrigation water, precipitation                                                      | 1123                  |
| S2     | 1434          | 102°50'E, 38°21'N   | Vegetation xylem (Salix matsudana,  Haloxylon ammodendron), 0-100 cm  soil, precipitation                       | 237                   |
| S3     | 1443          | 102°45'E, 38°13'N   | Vegetation xylem (Salix matsudana,  Haloxylon ammodendron, Phragmites  australis), 0-100 cm soil, precipitation | 205                   |
| S4     | 1467          | 102°42'E, 38°06'N   | Phragmites australis xylem,                                                                                     | 76                    |

**precipitation**

All precipitation samples from the sampling points were collected using rain gauges with a diameter of 20 cm. After each precipitation event, the liquid precipitation samples were immediately transferred to 100 mL sample bottles and stored at low temperatures. For soil samples, a 1 m soil auger was used to obtain samples, with layer sampling at intervals of 10 cm. In the downstream farmland area of the basin (S1), the collected irrigation water was sourced from observation wells near the sampling points. The collection of soil samples in the farmland area was influenced by irrigation events; during the irrigation season, samples were taken once before irrigation and then continuously for five days after irrigation. In the non-irrigation season, soil samples were collected once every seven days. Soil samples were placed in 50 mL glass bottles and stored frozen. All irrigation water samples were taken from observation wells near the sampling points.

We selected healthy woody and herbaceous plants, cutting side branches of *Salix matsudana* and *Haloxylon ammodendron*, as well as the non-green parts at the junction of the rhizomes of corn and *Phragmites australis*. These samples were stripped of their bark, placed in 50 mL glass bottles, and stored frozen. Automatic weather stations were set up near each sampling point to collect basic meteorological data including temperature, relative humidity, and atmospheric pressure during the study period. Additionally, daily precipitation and evaporation data from the Wuwei Shiyang River Experimental Station were collected for the period from January 1, 2020, to December 31, 2021."

**Comment five:**

**"Discussion:**

Critical Pathway: The proposed sequence of "enhanced transpiration  $\rightarrow$  suppressed sub-cloud re-evaporation  $\rightarrow$  increased precipitation" relies on the intermediary pathway of "near-surface humidity and temperature  $\rightarrow$  LCL  $\rightarrow$  raindrop re-evaporation rate." I recommend strengthening this argument by establishing clear statistical associations between observational/reanalysis data for relative humidity (RH), LCL, and fre-ev>."

**Response:** Your suggestion is very reasonable. In Section 5.2, "Regulation of Water Vapor Saturation by Vegetation and Atmosphere," we have reorganized the relationship between vegetation transpiration and the secondary evaporation under precipitation clouds according to the logical relationship of near-surface humidity and temperature  $\rightarrow$  LCL  $\rightarrow$  raindrop re-evaporation rate. The details of the modifications are as follows:

"The amount of sub-cloud evaporation loss of raindrops is mainly determined by relative humidity, atmospheric temperature, and rainfall amount. Although croplands and forests have significantly higher average LCL values (Figure 6), grasslands and shrubs have higher relative humidity. Therefore, more water vapor from raindrop evaporation converges in the atmosphere above grasslands and shrubs. To ensure heat and moisture balance in the lower atmosphere, there is a clear negative feedback between vegetation and atmosphere, meaning that simultaneous increases in vegetation evapotranspiration and

raindrop re-evaporation ratios will not occur in the atmosphere. As evapotranspiration increases, vertical water vapor recycling gradually accelerates, with evaporative cooling effects driving changes in precipitation and temperature at the microclimate scale."

"Impact of Irrigation: For oasis croplands, agricultural irrigation is the primary water source. The authors must explicitly address how irrigation water, as distinct from natural precipitation, influences the calculated cropland evapotranspiration fluxes and the subsequent interpretation of the local water vapor recycling."

**Response:** Thank you for your reminder; this is indeed our oversight. We have supplemented the irrigation water data and, in conjunction with our team's previous relevant research, clarified the impact of irrigation events on corn evapotranspiration and the recycling of water vapor. Additionally, we have included the collection times and frequency of the irrigation water in Section 3.1: Sample Collection. Moving forward, we will discuss the following topics and incorporate them into Sections 5.1 and 5.2:

**1. The Evaporative Fractionation Characteristics of Irrigation Water**

Irrigation water is one of the primary sources for vegetation growth in oasis farmland in arid regions. By analyzing the stable isotope composition characteristics of irrigation water, precipitation, and shallow soil water (0-10 cm) collected during the 2019 and 2021 growing seasons, we can fit a  $\delta^2$ H- $\delta^{18}$ O plot. Through this fitting of the water line, we can analyze the evaporative fractionation characteristics of irrigation water. The results indicate that:

Compared to the isotopes in precipitation, the isotopes in irrigation water are relatively enriched, showing a lower degree of evaporative fractionation. In contrast, the isotope values of soil water are lower, suggesting a higher intensity of evaporative fractionation. This also implies that irrigation water is directly introduced into the soil, serving as a stable water source for replenishing soil moisture and supporting plant growth.

**2. The Impact of Irrigation Water on Ecosystem Evapotranspiration and Recycled Water Vapor**

First, analyzing the evaporative fractionation characteristics and recharge relationships of irrigation water and soil water indicates that irrigation water directly infiltrates the soil in spring (Figure S1). Second, the water used for irrigation in farmlands is primarily concentrated in the dry season, specifically in April and May. However, during this period, regional temperatures and relative humidity are both relatively low, inhibiting evaporation losses from the soil. This suggests that agricultural irrigation mainly influences the evapotranspiration and recycled water vapor of corn fields by altering the intensity of soil evaporation in spring.

Existing studies have shown that based on  $\delta^2$ H, the estimated evaporation ratio of shallow soil (FE) during spring, dominated by irrigation water, has an average value of 39.59%, while the average FE during summer, dominated by precipitation, is 57.98% (Jiao et al., 2023). Combined with the analysis in this study, we conclude that the influence of irrigation water on the evapotranspiration (ET) and recycled water vapor ( $f_{re}$ ) of farmlands throughout the corn growing season is limited. Instead, the high

temperatures and precipitation dominating summer soil evaporation are the primary factors affecting the intensity of ET and  $f_{re}$  in the agricultural ecosystem (Zhu et al., 2019).

Figure S1: distribution and fitting characteristics of  $\delta^2$ H- $\delta^{18}$ O in precipitation, irrigation water, and soil water.

**References**

Jiao Y, Zhu G, Meng G, et al. Estimating non-productive water loss in irrigated farmland in arid oasis regions: Based on stable isotope data[J]. Agricultural Water Management, 2023, 289: 108515.

Zhu G, Guo H, Qin D, et al. Contribution of recycled moisture to precipitation in the monsoon marginal zone: Estimate based on stable isotope data[J]. Journal of Hydrology, 2019, 569: 423-435.

**Responses to the minor comments**

L11: Revise "four different vegetation cover areas" to "four vegetation cover types".

**Response:** We have revised this sentence in the original text.

L14: Express the range as "60–70%" (using an en-dash).

**Response:** Thank you for your reminder. We have corrected this error.

L15: Revise to: "...further suggesting that enhanced vegetation transpiration may increase precipitation by reducing below-cloud re-evaporation losses."

L26: The citation should be: (Huang et al., 2023).

**Response:** We have carefully checked the citation formats for all references in the manuscript and have corrected them.

L38: Revise to: "As a relatively humid enclave at the desert margin..." to avoid mischaracterizing an

oasis as a broadly humid region.

**Response:** We have revised this sentence according to your suggestions.

*L67: Please verify the figure citation; it is likely intended to be (Fig. 1).*

**Response:** We will carefully check the citation formats for all images and tables in the manuscript. Thank you very much for your reminder.

L71: Cite the dataset mentioned in the Acknowledgements here: "Daily precipitation and evaporation data were provided by the National Ecosystem Science Data Center, National Science & Technology Infrastructure of China."

**Response:** We have added the citation for this dataset in the acknowledgments section.

L117: Discuss the potential uncertainty introduced by using surface atmospheric pressure as a proxy for water vapor in the calculation of the F value.

**Response:** Through a review of a substantial amount of literature, we found that the calculation of the F value using surface atmospheric pressure is primarily due to the positive correlation between the water vapor content in the study area and the surface atmospheric pressure. Additionally, since it is difficult to obtain the values of initial and final water vapor content, we ultimately decided to use surface atmospheric pressure for this determination. In the explanation of Equation (10), we will supplement the following content:

"Typically expressed as the ratio of final to initial water vapor, the parameter F primarily reflects the atmospheric moisture conditions that affect precipitation formation in the region. To determine the value of F, we utilized the surface vapor pressure from each site, as water vapor content is positively correlated with the surface vapor pressure throughout the entire study area (c=1.657e, where c is the water vapor content in mm, e is the surface vapor pressure in hPa, R2 200 =0.94)(Hu et al., 2015)."

**References**

Hu W, Yao J, He Q, et al. Spatial and temporal variability of water vapor content during 1961–2011 in Tianshan Mountains, China[J]. Journal of Mountain Science, 2015, 12: 571-581.

L128 & L133: Ensure consistency between in-text citations and the reference list. The main text cites "Pruppacher and Klett, 2010" and "Rogers and Yau, 1989," but these must match the entries in the References section.

**Response:** We have changed it to "Pruppacher and Klett, 1978" and "Rogers, 1976."

L223: Correct the numbering to "5 Discussion".

L273: Correct the subheading to "6 Conclusion".

**Response:** We have corrected the error in the title numbering.

*Table 1: Please add a footnote explaining the meaning of the "/" symbol.*

**Response:** Thank you for your reminder. The corrected statement is as follows:

**Table 2:** Variations in stable isotopes of precipitation, soil water, and xylem water in oasis vegetation (/ indicates missing values).

Figure 1: The caption should clearly differentiate the panels: "(a) represents the sampling point..., (b) represents..., (c) represents..."

**Response:** Thank you for your reminder. We have modified the title of Figure 1, and the details of the changes are as follows:

**Figure 1:** Overview of the study area. (a) represents the sampling point locations and vegetation types, (b) represents the geographic location of the study area, (c) represents the complementary relationship between recycled water vapor and secondary evaporation vapor under rain clouds (from Google Maps).

Figures 2 & 4: Perform statistical significance tests on the data presented and indicate the corresponding p-values directly on the figures. In Figure 4, ensure the colors representing different rainfall intensities are consistent across panels (b), (c), and (d).

**Response:** For Figures 2, we conducted an overall test using the Kruskal-Wallis method and performed pairwise Mann-Whitney U tests, obtaining the final p-values after applying the Bonferroni correction. The modified figures is as follows:

**Figure 2:** Evapotranspiration capacity of different vegetation ecosystems. a represents the evapotranspiration composition ( $\delta_{ET}$ ) and transpiration ratio ( $f_T$ ) of the ecosystem, b represents the oxygen isotope composition ( $\delta_E$ ,  $\delta_T$ ) of soil evaporation and vegetation transpiration.

**For Figures 4:**

**Figure 4:** Sub-cloud secondary evaporation loss of raindrops. a represents the lifting condensation level of raindrops, b represents the re-evaporation rate of raindrops, c represents the change in  $\delta^{18}$ O value during raindrop fall, d represents the diameter of raindrops after evaporation.

Figure 7: In the caption, clarify panel (b) by stating: "Initial raindrop diameter back-calculated from the microphysical end-state based on the re-evaporation model..." to avoid confusion with panel (a).

**Response:** Thank you for your valuable feedback. We appreciate your suggestion regarding the clarification of panel (b) in Figure 7.

We used the Microphysical Model to back-calculate the initial diameter of raindrops. Based on your suggestion, we have revised the caption for panel (b) to read:" b represents the initial and final diameters of raindrops back-calculated from the Microphysical Model and the re-evaporation model." We hope this addresses your concern, and we thank you once again for your insightful suggestions.